# Growth Arrest of Alveolar Cells in Response to Cytokines from Spike S1-Activated Macrophages: Role of IFN-γ

**DOI:** 10.3390/biomedicines10123085

**Published:** 2022-12-01

**Authors:** Amelia Barilli, Rossana Visigalli, Francesca Ferrari, Giulia Recchia Luciani, Maurizio Soli, Valeria Dall’Asta, Bianca Maria Rotoli

**Affiliations:** 1Laboratory of General Pathology, Department of Medicine and Surgery, University of Parma, 43125 Parma, Italy; 2Immunohematology and Transfusion Medicine, University Hospital of Parma, 43125 Parma, Italy

**Keywords:** alveolar epithelial damage, autophagy, baricitinib, human macrophages, IFN-γ, proliferative arrest

## Abstract

Acute respiratory distress syndrome (ARDS) is characterized by severe hypoxemia and high-permeability pulmonary edema. A hallmark of the disease is the presence of lung inflammation with features of diffuse alveolar damage. The molecular pathogenetic mechanisms of COVID-19-associated ARDS (CARDS), secondary to SARS-CoV-2 infection, are still not fully understood. Here, we investigate the effects of a cytokine-enriched conditioned medium from Spike S1-activated macrophage on alveolar epithelial A549 cells in terms of cell proliferation, induction of autophagy, and expression of genes related to protein degradation. The protective effect of baricitinib, employed as an inhibitor of JAK-STAT, has been also tested. The results obtained indicate that A549 exhibits profound changes in cell morphology associated to a proliferative arrest in the G0/G1 phase. Other alterations occur, such as a blockade of protein synthesis and the activation of autophagy, along with an increase of the intracellular amino acids content, which is likely ascribable to the activation of protein degradation. These changes correlate to the induction of IFN-regulatory factor 1 (IRF-1) due to an increased secretion of IFN-γ in the conditioned medium from S1-activated macrophages. The addition of baricitinib prevents the observed effects. In conclusion, our findings suggest that the IFN-γ-IRF-1 signaling pathway may play a role in the alveolar epithelial damage observed in COVID-19-related ARDS.

## 1. Introduction

In COVID-19, about one third of hospitalized patients develops acute respiratory distress syndrome (ARDS) [1,2], a life-threatening inflammatory lung condition that typically manifests with severe hypoxemia and high-permeability pulmonary oedema [3]. A hallmark of ARDS is the presence of lung inflammation with features of diffuse alveolar damage (DAD): a massive death of epithelial and endothelial cells occurs during the “exudative” phase of the syndrome, followed by a “proliferative” phase characterized by alveolar type II cell hyperplasia and interstitial fibrosis [4]. Although it is recognized that COVID-19-associated ARDS (CARDS) is characterized by a higher risk of coagulation and thromboembolic complications than classical ARDS [5,6], the pathogenetic mechanisms underlying the onset of this syndrome are not yet fully understood [7,8]. A prominent role is, however, ascribed to the massive release of cytokines within an exaggerated inflammatory response to SARS-CoV-2 virus, the so called “cytokine storm” [9]; elevated serum levels of cytokines and chemokines have been described in patients with COVID-19 and they correlate with the severity of the disease [10,11].

Among the inflammatory markers involved, a pivotal role in the onset of COVID-19 disease is played by dysregulated interferon responses [12]. Interferons (IFNs) are pleiotropic cytokines that mediate antiviral, anti-proliferative, anti-tumor and immuno-modulatory activities. By binding to different receptors these cytokines activate downstream pathways based on Janus kinases (JAKs) and STAT signaling [13]. Among the different subtypes of interferons, type I interferons (mainly IFNα and β) preferentially trigger the formation of the ISG factor 3 (ISGF3) complex composed of phosphorylated STAT1 and STAT2 together with IFN-regulatory factor 9 (IRF-9) [14]. Type II interferon, (IFN-γ) instead, induces the phosphorylation of STAT1 that results in the formation of STAT1 homodimers that induce IRF-1 expression [15]. This latter step is required for the delayed transcription of many secondary IFN-γ-induced genes [15]. IRF-1, constitutively expressed at a very low level, is dramatically transcribed upon stimulation by cytokines, infections and IFNs. IRF-1 has been shown to stimulate the expression of various genes with anti-viral and anti-proliferative effects [16] that range from alterations in lipid metabolism to the activation of protein degradation machinery [17].

In our previous study we demonstrated that the treatment of human macrophages with SARS-CoV-2 spike-S1 protein induced a high secretion of cytokines that, in turn, activated alveolar epithelial cells. In particular, the exposure of human alveolar A549 cells to supernatants from S1-treated macrophages, by activating JAK/STAT pathways, caused a further release of inflammatory mediators and a dysfunction of epithelial barrier integrity [18]. Here, we aim to deepen the effects of cytokine-enriched conditioned medium from S1-activated macrophages on alveolar A549 cells; to this end, the effects on cell proliferation, the induction of autophagy and the expression of genes related to protein degradation have been evaluated and correlated with the induction of IRF1. The protective effect of baricitinib, employed as an inhibitor of JAK-STAT, has also been tested.

## 2. Materials and Methods

### 2.1. Cell Models

Alveolar carcinoma A549 cells, obtained from the American Type Culture Collection (ATCC, Rockville, MD, USA) were cultured in RPMI1640 medium supplemented with 10% FBS and 1% penicillin/streptomycin at 37 °C in a humidified atmosphere with 5% CO_2_. For experiments, cells were seeded at a density of 25 × 10^3^ cells/well in 48-well culture plates or 1 × 10^5^ cells/well in 12-well culture plates.

### 2.2. Experimental Treatments

A549 cells were treated with Conditioned Media (CM) collected from monocyte-derived macrophages (MDM), obtained as already described [18,19]. To this end, MDM were incubated in the absence and in the presence of 5 nM S1 subunit of SARS-CoV-2 spike recombinant protein (ARG70218; Arigo Biolaboratories, Hsinchu City, Taiwan) premixed with 2 µg/mL Polymyxin B, to exclude any possible contamination by lipopolysaccharides (LPS). After 24 h, the culture media of macrophages were collected as conditioned medium from control MDM (CM_cont) and S1-treated MDM (CM_S1). The media obtained from MDM of 16 different donors were pooled and employed for the treatment of A549 cells. Where indicated, these cells were pre-treated with 1 µM baricitinib for 2 h before the addition of CM_S1 and the inhibitor left in the culture medium throughout the experiment.

### 2.3. Cell Proliferation and Viability Assay

A549 cells were plated at a density of 50 × 10^3^ cells/mL in 48-well plates and cultured overnight before the treatment with conditioned medium from control MDM (CM_cont) or S1-treated MDM (CM_S1) for 24, 48 and 72 h. Proliferation was assessed by counting the number of adherent cells; to this end, monolayers were washed twice in PBS and cells were detached through trypsinization, then counted with a Cell Counter ZM (Coulter Electronics Ltd., Luton, UK). The cell viability was assessed in parallel by employing the resazurin method [20]. According to this method, viable cells reduce the non-fluorescent compound resazurin into the fluorescent resorufin that accumulates into the medium. After the treatments, A549 cells were thus incubated for 1 h with fresh growth medium supplemented with 44 μM resazurin and fluorescence was then measured at 572 nm with a fluorimeter (EnSpire Multimode Plate Readers; PerkinElmer, Monza, Italy).

### 2.4. Cell Cycle Analysis

For the analysis of the cell cycle, A549 cells were seeded at a density of 10 × 10^5^ cells/mL in 12-well plates. The day after, cells were treated with CM_cont or CM_S1 for 24 h, then harvested through trypsinization, and incubated for 18 h at 4 °C in a hypotonic solution containing 50 μg/mL propidium iodide (PI) and 10 μg/mL RNaseA. Cell cycle distribution was determined using a FACScan flow cytometer (FACSCalibur; BD Biosciences, Franklin Lakes, NJ, USA); resulting data were analyzed with Kaluza Analysis Software (Beckman Coulter, Milano, Italy).

### 2.5. Autophagy Detection

Autophagy was determined in A549 cells grown in 96-well plates by employing a CYTO-ID^®^ Autophagy Detection Kit, as specified by the manufacturer’s instructions (ENZO Lyfe Sciences, Euroclone, Milano, Italy).

### 2.6. Determination of the Intracellular Amino Acid Content

The intracellular content of amino acids was determined with high performance liquid chromatography employing a Biochrom 20 amino acid analyzer (Biochrom, Cambridge, UK), as previously described [21]. Briefly, A549 cells, grown on 12-well trays, were incubated for 24 h with CM_cont or CM_S1 and the intracellular pool was extracted with a 10-min incubation in ethanol at 4 °C. Samples were lyophilized and re-suspended in 150 μL Lithium Loading Buffer; the intracellular content of each amino acid species was then determined employing a high-resolution lithium column and lithium buffers for elution (Biochrom). The column effluent was mixed with an EZ Nin Reagent Kit (Biochrom), passed through the high-temperature reaction coil, and read by the photometer unit at both 570 and 440 nm. The protein content in each condition was determined using a modified Lowry procedure [22], and the content of amino acids was expressed as nmol/mg of protein.

### 2.7. Amino Acid Uptake

Amino acid uptake was determined in A549 cells seeded onto 96-well trays as previously described [23]. Briefly, after two rapid washes in a pre-warmed transport buffer [Earle’s Balanced Salt Solution (EBSS) containing (in mM) 117 NaCl, 1.8 CaCl_2_, 5.3 KCl, 0.9 NaH_2_PO_4_, 0.8 MgSO_4_, 5.5 glucose, 26 Tris/HCl, adjusted to pH 7.4], the cells were incubated for 1 min in the same solution containing [^3^H]glutamic acid (0.1 mM, 3 μCi/mL), [^3^H]glutamine (0.1 mM, 2 μCi/mL) or [^3^H]proline (0.1 mM, 5 μCi/mL). The experiment was terminated by two rapid washes (<10 s) in an ice-cold 300 mM urea. The ethanol-soluble pool was extracted in 0.1 mL ethanol and radioactivity was measured with the MicroBeta^2®^ liquid scintillation spectrometer (PerkinElmer, Milano, Italy). The monolayers were then dissolved in 1 N NaOH containing 0.5% sodium deoxycholate and assayed for protein content by a modified Lowry procedure [22]. Amino acid uptake is expressed as nmol/mg of protein/min.

### 2.8. RT-qPCR Analysis

The analysis of gene expression was performed with RT-qPCR, as already described [23]. Briefly, 1 µg of total RNA was reverse transcribed with a RevertAid RT Reverse Transcription kit (Thermo Fisher Scientific, Monza, Italy) and 20 ng of cDNA underwent qPCR on a StepOnePlus Real-Time PCR System (Thermo Fisher Scientific). Forward/reverse primer pairs detailed in Table 1 were employed for qPCR analysis along with PowerUp Sybr™ Green Master Mix (Thermo Fisher Scientific); the expression of the genes of interest was normalized to that of the housekeeping gene (*RPL15*) and calculated relative to its expression level in control conditions (=1), as specified in each Figure.

### 2.9. Cytokine Analysis

Cell culture supernatants from untreated (CM_cont) or S1-treated (CM_S1) monocyte-derived macrophages (MDM) were collected, centrifuged at 300× *g* for 10 min to remove particulates, and stored at −20 °C. The quantification of Interferon γ (IFN-γ) released in these media was then performed with the Human IFN-gamma Quantikine ELISA Kit (R&D Systems, Bio-techne, Milano, Italy), according to the manufacturer’s instructions. The concentrations of IFN-γ are given as pg/mL.

### 2.10. Western Blot Analysis

The analysis of protein expression was performed on cell lysates obtained with LDS sample buffer (Thermo Fisher Scientific), as already described [21]. 20 µg of proteins were separated in Bolt™ 4–12% Bis-Tris mini protein gel (Thermo Fisher Scientific) and transferred to PVDF membranes (Immobilon-P membrane, Thermo Fisher Scientific). Membranes were incubated for 1 h at RT in a blocking solution (4% non-fat dried milk in TBST, Tris-buffered saline solution +0.5% Tween), then overnight at 4 °C with anti-IRF-1 rabbit polyclonal antibody (1:2000, Cell Signaling Technology, Euroclone, Milano, Italy) in TBST containing 5% BSA. Anti-vinculin mouse monoclonal antibody (1:2000, Merck) was used as loading control. Horseradish peroxidase (HRP)-conjugated secondary antibodies (anti-rabbit and anti-mouse IgG, Cell Signaling Technology) were employed (1:10,000). Immunoreactivity was visualized with SuperSignal™ West Pico PLUS Chemiluminescent HRP Substrate (Thermo Fisher Scientific). Western Blot images were captured with an iBright FL1500 Imaging System (Thermo Fisher Scientific) and analyzed with iBright Analysis Software (Thermo Fisher Scientific).

### 2.11. Statistical Analysis

GraphPad Prism 9 (GraphPad Software, San Diego, CA, USA) was used for statistical analysis. *p* values were calculated with one-way ANOVA for matched measures and the Holm-Sidak correction for multiple comparisons, or with a Student’s *t*-test for paired data, as specified in the legend of each Figure. *p*-values < 0.05 were considered statistically significant.

### 2.12. Materials

Endotoxin-free fetal bovine serum (South America origin; EU Approved) was purchased from Euroclone (Milano, Italy). ^3^H-L-leucine, ^3^H-L-glutamic acid, ^3^H-L-glutamine and ^3^H-L-proline were from PerkinElmer, Milano, Italy. Merck (Milano, Italy) was the source of baricitinib, as well as of all other chemicals, unless otherwise specified.

## 3. Results

Recently, we have shown that the exposure of alveolar A549 cells to conditioned medium (CM) from spike S1-activated macrophages causes the activation of the epithelial cells; the reason is the massive presence of cytokines released from treated macrophages that, in turn, provokes a further release of inflammatory mediators from epithelial cells [18,24]. Given the relevance of the epithelial alveolar damage in COVID-19 disease, we examine here whether this treatment causes, in addition, signs of cell injury. To this end, A549 were exposed up to 72 h to a pool of conditioned media collected from untreated macrophages (CM_cont) or from S1-activated macrophages (CM_S1). Results indicate that CM_S1 causes an evident change of morphology, as highlighted by images obtained with phase contrast microscopy (Figure 1A). Indeed, when maintained in the presence of conditioned medium collected from untreated macrophages (CM_cont), A549 cultures appear confluent and cells exhibit a polygonal morphology; instead, when treated with CM_S1, cells acquire a spindle-shaped morphology and look sparser in monolayers. Anyway, no sign of cell detachment or death is evident, suggesting a cytostatic, rather than a cytotoxic, effect under the experimental condition adopted. To explore this issue, we next addressed the effect of CMs on the proliferative status of alveolar epithelial cells. To this end, we first monitored cell proliferation and viability in cultures maintained in the presence of CM_cont or CM_S1 for different times. The results obtained, shown in Figure 1B, demonstrate that, while CM_cont has no effect as compared to cells maintained under normal growth conditions (RPMI), CM_S1 actually causes a severe growth arrest of A549 cells. Indeed, both the number (left) and the viability (right) of adherent cells exposed to this incubation medium do not increase with time. Consistently, results obtained with flow cytometry (Figure 1C) confirm that the percentage of cells in G0/G1, S and G2/M phases is comparable in the presence of CM_cont as in cells maintained in complete growth RPMI medium. Conversely, changes in cell distribution occur upon treatment with CM_S1, with a decrease in the number of cells in S phase (from 18.5% to 8.7%) and a concomitant increase of that in G0/G1 phase (from 66.3% to 81.2%); under the same condition, a significant decrease of global protein synthesis is also observed, as demonstrated by the significant reduction of ^3^H-leucine incorporation into nascent proteins (Figure 1D). Moreover, upon 72 h-incubation in CM_S1, A549 cells undergo a significant increase in autophagic activity, as compared to cells incubated in CM_cont (Figure 1E).

To better address the anti-proliferative effects of the conditioned medium from spike-activated macrophages, we next measured the intracellular pool of amino acids in treated A549 cells. As shown in Figure 2A, the incubation of epithelial cells with CM_S1 also causes a marked increase in the total amount of intracellular amino acids, with the overall content raising from 339 ± 43.1 to 551 ± 30.3 nmol/mg of protein. Under this condition, the intracellular content of all the amino acids, except aspartic acid, increases, with the major changes observed for glutamic acid and neutral amino acids such as glutamine, proline, glycine, and alanine. The greater amount of intracellular amino acids is not due to an increased uptake from the extracellular environment; indeed, the inwardly directed transport of glutamine and proline are comparable in CM_S1 and in CM_cont-treated cells, while that of glutamic acid even decreases (Figure 2B). This finding, by excluding an increased activity of transmembrane transporters, likely ascribes the increase of the intracellular amino acid pool observed in CM_S1-treated cells to the activation of protein degradation processes.

To further explore this issue, we next evaluated the pattern of expression of some markers of protein degradation pathways (Figure 3). In this context, the expression of genes related to the activity of the proteasome has been investigated, such as Ubiquitin D (UbD), which mediates proteasome-dependent protein degradation, Tripartite motif 22 (TRIM22), which is involved in the autophagy process, and the immunoproteasome subunit PSMB8. Results obtained indicate that the mRNA levels of all these genes significantly increase after 4 h of incubation with CM_S1; for TRIM22, an even much more marked induction is observed upon treatment for 24 h.

The transcriptional modulation of these genes is known to be under the control of IFN-regulatory factor 1 (IRF-1), which is typically induced by IFN-γ as part of the host antiviral response [17]. It is well known, indeed, that IFN-γ, through the binding to the IFNGR receptor, causes the activation of the JAK/STAT1 signaling pathway that ultimately results in the transcription of many genes, including IRF-1. To verify the activation of this signaling pathway under our experimental conditions, we first measured the amount of IFN-γ in the incubation media, which actually confirmed a higher concentration of the cytokine in CM_S1 than in CM_cont (Appendix A). Therefore, we next addressed the expression of IRF-1 transcription factor in A549 cells treated with conditioned media both in the absence and in the presence of baricitinib, a well-known FDA-approved inhibitor of the JAK/STAT pathway [25]. Similarly to UbD, TRIM22 and PSMB8, also the expression of IRF-1 is stimulated upon incubation with CM_S1 at both the gene and protein level, with a marked increase evident after 4 h of treatment that declines after 24 h (Figure 4A). The addition of baricitinib to the incubation medium actually hinders the increase of IRF-1 mRNA and protein, thus indicating the activation of the JAK/STAT-IRF1 axis upon the incubation of alveolar cells with CM_S1. The drug also prevents the CM_S1-dependent induction of genes related to proteasome-dependent protein degradation (Figure 4B), as well as the activation of autophagy (Figure 4C); consistently, in the presence of the drug, the changes observed in the morphology of A549 exposed to CM_S1 are no more evident, and the count of the cell number is significantly increased compared to CM_S1-treated cells (Figure 4D). Overall, these findings suggest a potential role for the JAK/STAT pathway, via IRF-1-dependent mechanisms, in mediating the cytostatic effects on alveolar epithelial cells of the cytokines secreted by S1-activated macrophages, and sustain the protective effect of baricitinib.

## 4. Discussion

The hallmark of COVID-19-associated ARDS (CARDS) is the diffuse alveolar damage (DAD) characterized by the massive death of epithelial and endothelial cells due to the excessive secretion of inflammatory cytokines (the so-called “cytokine storm”) by innate immune cells and epithelial cells [9]. In this context, a prominent role is ascribable to macrophages, including resident alveolar macrophages (AMs) and transient monocytes/macrophages recruited from the blood [10].

Recently, we demonstrated that the exposure of macrophages to SARS-CoV-2 spike S1 induces the production of a great amount of cytokines and chemokines, mainly IL-6 and IL-8 [19], which act on alveolar epithelial cells causing a further secretion of inflammatory mediators, as well as the impairment of the epithelial barrier integrity through the activation of the JAK/STAT pathway [18]. Here, by further addressing the effects of the cytokine-enriched conditioned medium from S1-activated macrophages on A549 alveolar cells, we observed profound changes in epithelial cell morphology associated to a proliferative arrest in the G0/G1 phase. Under the same conditions, other alterations have been found, such as the blockade of protein synthesis and the induction of autophagy, along with an increase of the intracellular amino acids content, likely ascribable to the activation of protein degradation.

Consistently, CM_S1 also causes an increased expression of genes that are correlated to autophagy and Ubiquitin-Proteasome System, the two major intracellular mechanisms strictly interconnected for protein quality control [26]. More precisely, a marked induction of the proteasome degradation signal UBD (also known as FAT10) [27] has been observed, along with an up-regulation of the proteasome subunit beta type-8 PSMB8. An even greater induction of TRIM22, an interferon-inducible protein that augments autophagy in different cells [28,29], is also evident.

Cell fate upon activation of autophagy under our experimental conditions remains to be elucidated. Autophagy is recognized as a survival mechanism that helps defending organisms against degenerative, inflammatory, neoplastic, and infectious diseases. Also, the infection with SARS-CoV-2 pseudovirions has been shown to activate autophagy in three different cell models [30]; there, however, pro-apoptotic responses were ultimately observed. Actually, it is recognized that an excessive activation of autophagy may lead to cell death. Under our experimental conditions, conversely, the induction of autophagy, at least until 72 h of incubation, is not associated to signs of cell death. The discrepancy between that and our study can be due to the different triggering stimuli adopted. In our hands, the induction of autophagy by CM_S1 in A549 cells is not caused by viral infection, but by a cytokine enriched conditioned medium obtained from spike-activated macrophages. Proinflammatory cytokines are actually another well-established autophagic stimulus: IFN-γ, in particular, has been described to augment autophagy that, in turn, stimulates the release of the cytokine in a positive feedback loop [31]. Therefore, although we cannot exclude the involvement of other mediators, it is likely to suppose that the effects we described are mostly ascribable to IFN-γ, whose amount is significantly higher in conditioned medium from S1- than from untreated cells. Indeed, the observed up-regulation of IRF-1, a transcription factor specifically induced by Type II IFNs [32], sustains a role for IFN-γ in the activation of autophagy under our experimental conditions. Among the many functions mediated by IRF-1, the overexpression of this factor is found to induce autophagy in hepatocarcinoma cells and its silencing blocks IFN-γ-mediated autophagy [33]. Importantly, an up-regulation of IRF-1 has been described in different tissues including lung and liver in critical COVID-19 disease [34].

The JAK/STAT pathway is currently recognized as a signalling mechanism central to the response and secretion of cytokines and chemokines in COVID-19 [24,35,36]; thus, it is now generally accepted that targeting JAKs represents a valid therapeutic strategy for the treatment of the disease [37]. Among the JAK-STAT inhibitors, baricitinib proved effective in preventing the progression to a severe form of COVID-19 with reduced hospitalization and mortality [38,39,40,41,42]. We have recently described that baricitinib in vitro limits the secretion of different cytokines and chemokines by macrophages and endothelial cells [24]; in particular, the release of IFN-γ-Induced Protein 10 (IP10), which depends upon the activity of IRF-1 [43,44], was completely abolished. Here we demonstrate that baricitinib is able to decrease the synthesis of IRF-1 in A549 cells and to eliminate almost completely the expression of its downstream targets TRIM22, UbD and PSMB8. Consistently, autophagy and the changes in cell morphology are completely prevented by the drug, and the anti-proliferative effects are partially reversed.

## 5. Conclusions

Overall, our data indicate that inflammatory mediators released by S1-activated macrophages exert, on alveolar epithelial cells, a cytostatic effect associated with the activation of autophagy. In particular, the axis IFN-γ-IRF-1 seems to play a central role in the induction of these effects which could ultimately be involved in the onset of the alveolar epithelial damage observed in COVID-associated ARDS.

## Figures and Tables

**Figure 1 biomedicines-10-03085-f001:**
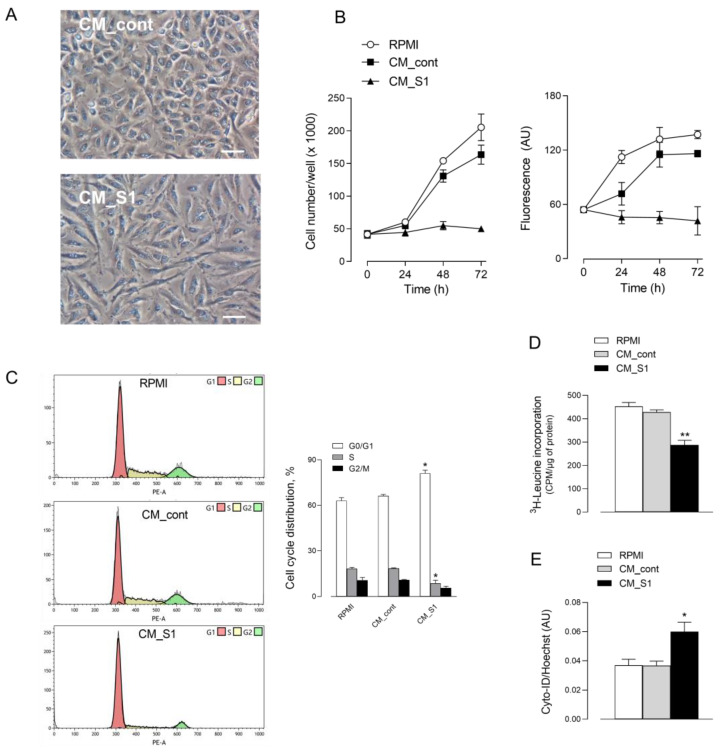
Effects of conditioned medium from S1-treated macrophages on the viability and proliferation of alveolar epithelial cells. A549 cells were maintained in RPMI1640 medium (RPMI) or incubated in conditioned medium obtained by incubating monocytes-derived macrophages in the absence (CM_cont) or in the presence of 5 nM S1 (CM_S1). (**A**) Phase contrast microscopy images of cells treated for 48 h. Bar = 100 µM. (**B**) Cell proliferation. At the indicated times, cell proliferation was assessed by counting the number of adherent cells (left), or by measuring cell viability through resazurin assay (right), as described in Methods. Each point represents the media ± SD of five determinations in a representative experiment that, repeated three times, gave comparable results. (**C**) Cell cycle analysis. After 24 h of incubation, cells were harvested, stained with propidium iodide, and analyzed for cell cycle with flow cytometry as detailed in Methods. Plots obtained in a representative experiment are shown (left). Cell distribution at each phase of cell cycle is also shown (right); bars represent the mean ± SEM of data obtained in three independent experiments. * *p* < 0.05 vs. CM_cont with ANOVA. (**D**) Protein synthesis. After 24 h, protein synthesis was determined by evaluating the incorporation of ^3^H-leucine, as described in Methods. Bars represent the mean ± SEM of three independent experiments. ** *p* < 0.01 vs. CM_cont with ANOVA. (**E**) Induction of autophagy. Autophagy was determined after 72 h as described in Methods. Bars represent the mean ± SEM of four determinations. * *p* < 0.05 vs. CM_cont with ANOVA.

**Figure 2 biomedicines-10-03085-f002:**
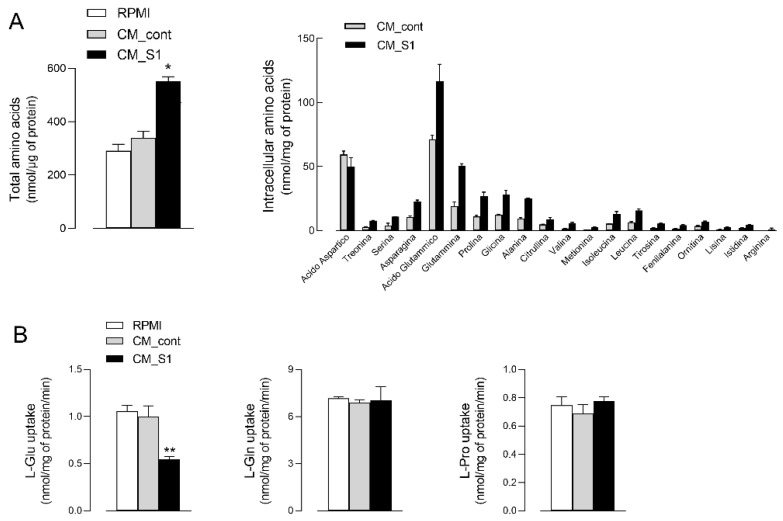
Effects of conditioned medium from S1-treated macrophages on the content and uptake of amino acids in alveolar epithelial cells. A549 cells were incubated for 24 h under the same conditions as in Figure 1. (**A**) Amino acid content. The intracellular pool of amino acids was analyzed as described in Methods. Total intracellular amino acid content (left) is calculated as the sum of the single indicated amino acids (right). Bars represent the mean ± SEM of three independent experiments. * *p* < 0.05 vs. CM_cont with ANOVA. (**B**) Amino acid transport. Uptake of ^3^H-L-glutamic acid, ^3^H-L-glutamine and ^3^H-L-proline was determined as described in Methods. Bars represent the mean ± SD of four independent determinations within a representative experiment that, repeated three times, gave comparable results. ** *p* < 0.01 vs. CM_cont with ANOVA.

**Figure 3 biomedicines-10-03085-f003:**
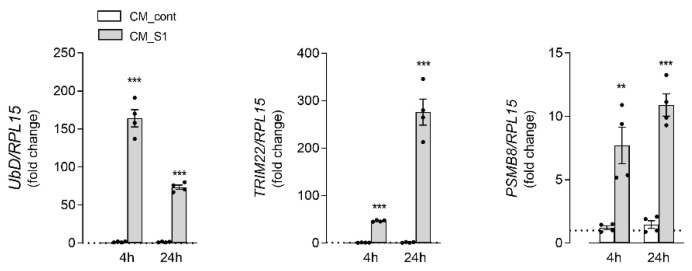
Effects of conditioned medium from S1-treated macrophages on the expression of proteasome-related proteins in alveolar epithelial cells. A549 cells were incubated under the same conditions as in Figure 1. The expression of the indicated genes was measured after 4 or 24 h of incubation by means of RT-qPCR, as described in Methods. The expression of each gene is shown as fold change of the mRNA measured in cells maintained in RPMI (=1, dotted line). Bars are means ± SEM of four independent experiments (single dots), each performed in duplicate. ** *p* < 0.01, *** *p* < 0.001 vs. CM_cont with Student’s *t*-test.

**Figure 4 biomedicines-10-03085-f004:**
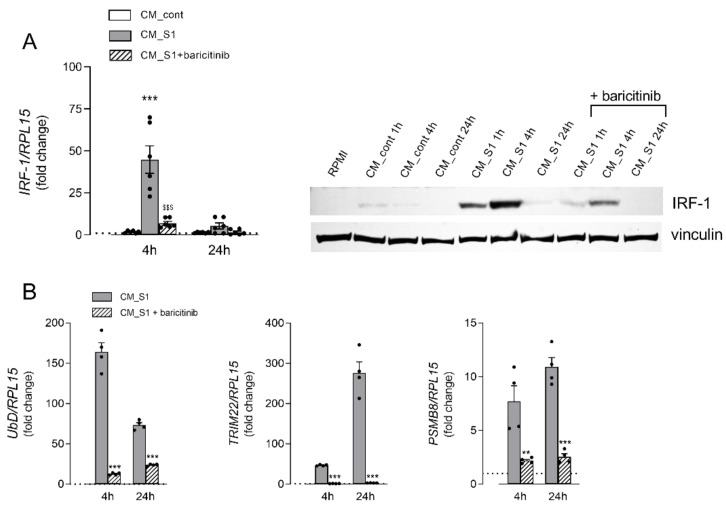
Effects of baricitinib on the cytostatic effects of conditioned medium from S1-treated macrophages in alveolar epithelial cells. A549 cells were maintained in RPMI or incubated in CM_cont or CM_S1 for the different times; where indicated, 1 µM baricitinib was added to CM_S1, as detailed in Methods. (**A**) IRF-1 expression. The expression of IRF-1 mRNA (left) was measured by means of RT-qPCR and expressed as the fold change of cells maintained in RPMI (=1, dotted line). Bars are means ± SEM of four independent experiments (single dots), each performed in duplicate. *** *p* < 0.001 vs. CM_cont; ^$$$^ *p* < 0.001 vs. CM_S1 with ANOVA. The expression of IRF-1 protein was assessed by means of western blot analysis (right), as detailed in Methods. A representative blot is shown that, repeated three times, gave comparable results. (**B**) Gene expression. The expression of the indicated genes was measured after 4 or 24 h of incubation by means of RT-qPCR, as described in Methods. The expression levels of each gene are expressed as the fold change of cells maintained in RPMI (=1, dotted line). Bars are means ± SEM of four independent experiments (single dots), each performed in duplicate. ** *p* < 0.01, *** *p* < 0.001 vs. CM_S1 with Student’s *t*-test. (**C**) Autophagy. After 72 h, autophagy was determined by employing CYTO-ID^®^ Autophagy Detection Kit, as described in Methods. Each bar represents the mean ± SD of five determinations within a representative experiment that, repeated three times, gave comparable results. ** *p* < 0.01 vs. CM_cont; ^$^ *p* < 0.05 vs. CM_S1 with ANOVA. (**D**) Cell proliferation. Cells were counted after 48 h treatment, as described in Methods. Each bar represents the mean ± SD of five determinations within a representative experiment that, repeated three times, gave comparable results. *** *p* < 0.001 vs. CM_cont; ^$$$^ *p* < 0.001 vs. CM_S1 with ANOVA. (**E**) Phase contrast microscopy images of A549 cells treated for 72 h. Bar = 100 µM.

**Table 1 biomedicines-10-03085-t001:** Sequence of primer pairs employed for RT-qPCR analysis.

Gene (Gene ID)	Forward Primer	Reverse Primer
*RPL15* (6138)	GCAGCCATCAGGTAAGCCAAG	AGCGGACCCTCAGAAGAAAGC
*IRF1* (3659)	CTGTGCGAGTGTACCGGATG	ATCCCCACATGACTTCCTCTT
*PSMB8* (5696)	CACGCTCGCCTTCAAGTTC	AGGCACTAATGTAGGACCCAG
*TRIM22* (10346)	ACCAAACATTCCGCATAAACGA	AGGCGGTTCTCTCTTGTCTGA
*UBD* (10537)	GAAGCCTCTCATCTTATGGCATT	CCTCATCACCTGACTCCACAA

## Data Availability

Data are contained within the text; original blots are also supplied.

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
