# Peer review of "Growth Arrest of Alveolar Cells in Response to Cytokines from Spike S1-Activated Macrophages: Role of IFN-γ"

_biomedicines, 2022, doi:10.3390/biomedicines10123085_

Round 1

Reviewer 1 Report

Acute respiratory distress syndrome (ARDS) is characterized by severe hypoxemia and high permeability pulmonary edema. The hallmark of the disease is the presence of lung inflammation with characteristics of diffuse alveolar damage (DAD). The authors of this manuscript indicate that inflammatory mediators released by S1-activated macrophages exert, on alveolar epithelial cells, a cytostatic effect associated with an activation of autophagy. In particular, the axis IFN-IRF-1 seems to play a central role in the induction of these effects, which could ultimately account for alveolar epithelial damage observed in COVID-associated ARDS.

The manuscript is very well presented and the results are supported by an excellent discussion.

I suggest a little revision in writing. Well, there are small errors that can be easily corrected with a quick review.

Anyway, I suggest the approval of this manuscript for publication in the journal Biomedicines

Author Response

We thank the Reviewer for his/her comments. According to his/her suggestions, the text has been completely revised so as to eliminate spelling and typing errors.

Reviewer 2 Report

There are several critical points in this paper that cannot be corrected in a short period of time.

 First, the authors' evidence of effects on S protein action is insufficient. In order to really prove the inhibition of cell cycle arrest by S protein, it is essential to design an experimental group of S protein and a control group that suppresses the effect of S protein. For this, the authors had to use neutralizing antibodies to the S protein.

Second, cell-level signals cannot fully explain the pathology of the entire corona infection. To make such a claim, the authors should have identified differences in the in vivo effects of S protein in IRF-1 dropout mutant mice and normal mice.

It may be difficult for these additional experiments to end within a few months. However, it is clear that this reinforcement of evidence will make the authors' argument clearer.

Author Response

We would like to thank the Reviewer for his/her collaborative criticisms.

Regarding the first question, we believe that the description of our findings was probably misleading: the growth arrest that we observed in epithelial cells is not due to S1 but to the pool of cytokines produced by macrophages in response to the viral protein. This issue is now better explained in the final section of the introduction and throughout the text. The activation of immune cells in COVID-19 disease is nowadays recognized both in vivo, with the induction of the so called “cytokine storm”, and in vitro (among the different contributions on this issue, we have recently described this effect in human monocytes-derived macrophages in PMID: 34572407). According to our results (present and past), the presence of cytokines in the incubation medium is responsible for the induction of many effects in alveolar epithelial and endothelial cells; in particular, we demonstrate here that the activation of interferon γ-mediated pathways associates with the induction of growth arrest and autophagy in A549 cells. Given the inter-individual variability of cytokines production (for IFNg see Figure S1 in the present ms.; for other cytokines see previous contributions), we chose to use a pool of conditioned media obtained from macrophages from different donors. The manuscript has been revised, so as to better describe our findings and avoid any misunderstanding.

As for the second issue, we agree with the Reviewer that cell-level signals cannot fully explain the mechanisms of the entire corona infection, but we would like to underline that this is neither the aim of our contribution. As stated above, indeed, we demonstrate here that the presence of inflammatory cytokines due to the activation of macrophages by S1 protein activates the INFγ-IRF1 axis in alveolar epithelial cells and that this, in turn, associates with cell growth arrest; since the addition of baricitinib, by limiting the release of this and (we are aware) other inflammatory mediators, prevents/attenuates these effects, we propose that IFNγ is involved, although not exclusively, in the events described. Anyway, we agree with the Reviewer that our assumptions sometimes sounded too definitive; the ms. has been thus revised, so as to maintain more appropriate tones.

Reviewer 3 Report

Amelia Barilli and colleagues present a quality and well-written manuscript describing Growth arrest of alveolar cells in response to cytokines from spike S1-activated macrophages: role of IFN-gamma.

Authors investigated the effects of a cytokines-enriched conditioned medium from Spike S1-activated macrophage on alveolar epithelial A549 cells in terms of cell proliferation, induction of autophagy and expression of genes related to protein degradation. The protective effect of baricitinib, employed as inhibitor of JAK-STAT, has been also tested. 

Authors obtained the results indicating that A549 exhibit profound changes in cell morphology associated to a proliferative arrest in G0/G1 phase. Other alterations occur, such as a blockade of protein synthesis and the activation of autophagy, along with an increase of the intracellular amino acids content, likely ascribable to the activation of protein degradation. These changes correlate to the induction of IFN-regulatory factor 1 (IRF-1) due to an increased secretion of IFN-gamma in the conditioned medium from S1-activated macrophages. The addition of baricitinib prevents the observed effects. 

Finally, authors conclude that their findings suggest that IFN-gamma-IRF-1 signaling pathway may play a role in the alveolar epithelial damage ob- served in COVID-19-related ARDS.

Overall, the manuscript is valuable for the scientific community and should be accepted for publication after edits are made.

===========================

Other comments:

1) Please check for typos throughout the manuscript.

2) With regards to COVID-19 and viral infections – authors are kindly encouraged to cite the following article that describes novel cell-based approaches for the treatment of such diseases.
DOI: 10.3390/biomedicines9010059

Reviewer 4 Report

General comments:

The authors reported that inflammatory mediators released by S1-activated macrophages exert, on alveolar epithelial cells, a cytostatic effect associated to an activation of autophagy.

Major comments:

1. Why the “amino acid uptake” need to be detected? Please provide the concept or rationale.

2. Please provide the suitable titles for each figure.

3. Please provide the subtitle for (A) to (?). No label for figure alphabet was mentioned in figure legends.

Minor comments:

1. “p” value: p should be typed in italic font.

2. Table 1: Please remove the protein and protein names as this is a primer information for mRNA. Moreover, the accession number for each gene need to be provided in this Table.

Author Response

We would like to thank the Reviewer for the attention he/she paid to our manuscript. According to his/her comments, we added a title to each figure in the legends and labels for the panels; suggested corrections for “p-values” and Table 1 have been also performed. As far as amino acid uptake is concerned, the rationale for its analysis is now better described and discussed.

Round 2

Reviewer 2 Report

the data level of this work is very insufficient